# Post-Acute COVID-19 Sequelae in a Working Population at One Year Follow-Up: A Wide Range of Impacts from an Italian Sample

**DOI:** 10.3390/ijerph191711093

**Published:** 2022-09-05

**Authors:** Danilo Buonsenso, Maria Rosaria Gualano, Maria Francesca Rossi, Angelica Valz Gris, Leuconoe Grazia Sisti, Ivan Borrelli, Paolo Emilio Santoro, Antonio Tumminello, Carolina Gentili, Walter Malorni, Piero Valentini, Walter Ricciardi, Umberto Moscato

**Affiliations:** 1Department of Woman and Child Health and Public Health, Fondazione Policlinico Universitario A. Gemelli IRCCS, 00168 Roma, Italy; 2Center for Global Health Research and Studies, Università Cattolica del Sacro Cuore, Largo Francesco Vito 1, 00168 Rome, Italy; 3Department of Public Health Sciences and Paediatrics, University of Torino, 10124 Torino, Italy; 4Leadership in Medicine Research Center, Università Cattolica del Sacro Cuore, 20123 Rome, Italy; 5Department of Health Science and Public Health, Section of Occupational Health, Università Cattolica del Sacro Cuore, Largo Francesco Vito 1, 00168 Rome, Italy; 6Department of Health Science and Public Health, Section of Hygiene, Università Cattolica del Sacro Cuore, Largo Francesco Vito 1, 00168 Rome, Italy; 7Institute of Public Health, Section of Hygiene, Università Cattolica del Sacro Cuore, 00168 Rome, Italy; 8National Institute for Health, Migration and Poverty, 00153 Rome, Italy; 9Medical School, Università Cattolica del Sacro Cuore, Largo Francesco Vito 1, 00168 Rome, Italy

**Keywords:** COVID-19, long COVID-19, occupational medicine, employment status, smoking, workers

## Abstract

Long COVID-19 is a term used to describe the symptomatic sequelae that develop after suffering from COVID-19. Very few studies have investigated the impact of COVID-19 sequelae on employment status. The aim of this research was to characterise sequelae of COVID-19 in a population of workers who tested positive for COVID-19, with a follow-up within one year of the acute illness, and to analyse the possible association between this and changes in the workers’ occupational status. In this retrospective cohort study, a questionnaire was administered to 155 workers; descriptive, univariate (chi-square tests), and multivariate (logistic regression model) analyses were carried out. The mean age was 46.48 years (SD ± 7.302); 76 participants were males (49.7%), and 33 participants reported being current smokers (21.3%). Overall, 19.0% of patients reported not feeling fully recovered at follow-up, and 13.7% reported a change in their job status after COVID-19. A change in occupational status was associated with being a smoker (OR 4.106, CI [1.406–11.990], *p* = 0.010); hospital stay was associated with age > 46 years in a statistically significant way (*p* = 0.025) and with not feeling fully recovered at follow-up (*p* = 0.003). A persistent worsening in anxiety was more common in women (*p* = 0.028). This study identifies smoking as a risk factor for workers not able to resume their job; furthermore, occupational physicians should monitor mental health more closely after COVID-19, particularly in female workers.

## 1. Introduction

In December 2019 the novel Severe Acute Respiratory Syndrome Coronavirus 2 (SARS-CoV-2) was discovered in Wuhan, China, and it was immediately associated with significant morbidity and mortality [1]. The SARS-CoV-2 infection can affect patients very differently, ranging from asymptomatic to critical illness, and involves various apparatuses causing many different symptoms including cough, fever, fatigue, loss of taste and smell, up to more severe conditions such as pneumonia [2]. The very rapid spread of the virus led the World Health Organization (WHO) to declare a global pandemic on 11 March 2021; since then, the COVID-19 pandemic has become a major public health emergency [3]. To date, over 500 million cases and 6 million deaths have been reported worldwide by the WHO [4].

Although the introduction of vaccines in January 2021 led to a significant reduction in the number of mild and severe symptomatic patients, the sequelae that COVID-19 can cause in the long term are still cause for concern [5,6]. Sequelae following COVID-19 are various and may include fatigue, muscle and/or joint pain, exercise intolerance, breathlessness, nausea, headaches, memory loss and mood disturbances [7,8]. The WHO uses the term “Post COVID-19 Condition” or “Long Covid” to identify the set of symptomatic sequelae that typically develop within three months of COVID-19 infection, last at least two months, and cannot be explained by alternative diagnoses [9].

The individual and societal impact of this condition has not been fully understood yet. Various studies have shown how this syndrome leads to significant limitations in functional abilities and to a general reduction in quality of life; this is the case, especially, though not exclusively, in patients who have experienced a severe disease or have undergone a hospital stay [10,11,12,13,14,15]. A longitudinal cohort study recently performed in China showed that the population surviving COVID-19 after a hospital stay, even after two years from the acute illness, had more lasting symptoms and more residual pain or discomfort, as well as anxiety or depression, compared to controls [16]. A survey conducted in Finland also showed that women report long-term symptoms more frequently and a lower quality of life compared to men [17].

Considering the scientific evidence highlighting that COVID-19 sequelae can reduce patients′ functional abilities, it needs to be established if these also have an impact on their occupational status; very few studies have investigated the impact of COVID-19 sequelae on employment, although these have shown important implications [18,19,20]. A retrospective cohort study conducted in Denmark highlighted that patients who underwent an hospital stay due to COVID-19 were less likely to have been returned to work three months after testing positive [18]. In an Australian longitudinal study, 71.3% of the patients undergoing a hospital stay reported persistent symptoms, and 11.4% did not return to work in the six months following acute illness due to poor health [19]. Another study conducted in Sweden highlighted that a substantial number of people had been on sick leave due to post COVID-19 condition, and that this was mainly associated with patients′ age, the severity of illness and having taken sick leave the year before [20].

The aim of this study is to evaluate the sequelae of COVID-19 in a population of workers who tested positive for COVID-19, with a follow-up within one year of the acute illness, and to analyse the possible association between COVID-19 sequelae and changes in the workers′ occupational status.

## 2. Materials and Methods

This is a retrospective cohort study; the study population included adult members of the households of children diagnosed with SARS-CoV-2 infection using reverse transcriptase polymerase chain reaction (RT-PCR) between 1 April 2020 and 31 April 2021, at the Department of Women and Child Health of the Fondazione Policlinico Universitario A. Gemelli IRCCS of Rome, Italy. The Institution is a regional referral COVID-19 centre for adults and also has a dedicated paediatric infectious disease inpatient unit and an outpatient paediatric post-COVID unit; in these units, the ISARIC survey has been used as a screening tool for persisting symptoms [21]. A sample of 155 patients was recruited amongst relatives of children affected by COVID-19 that have been evaluated in our centre (both children undergoing hospital stay and community patients assessed in the outpatient unit). Relatives who did not suffer from COVID-19, or were not in working age (18–65 years old) were excluded from the study. The included participants were administered a questionnaire [21] investigating possible sequelae of COVID-19, their health status in general and possible consequences of the illness on their occupational status. For unanswered questions the data were considered missing and those values were excluded from the statistical analyses.

### 2.1. Questionnaire

The questionnaire used in this study was developed for the ISARIC Global COVID-19 follow up protocol for adults [21,22] and associated standardized data-collection forms [23,24,25,26].

The initial page of the questionnaire explained the aim and purpose of the study to the participants, while the second page collected the participants’ consent for the study and for personal data processing. After that, the questionnaire was divided into three separate sections.

The first section of the questionnaire focused on socio-demographic data (age, gender, being a smoker, alcohol consumption), as well as data concerning the acute COVID-19 illness of the participants (COVID-19 diagnosis had to be investigated as the participants were relatives of affected children; therefore, if there was no primary COVID-19 infection, the participant was excluded from the study), if they underwent hospital stay, and if they reported any complications during primary infection.

A second section of the questionnaire investigated a change in the ability of the participants to perform daily tasks (to move, perform personal care, perform daily activities) or a change in health status (pain, anxiety, breathlessness, vision or hearing problems, lack of focus, ability to talk). It was also investigated if their ability to perform activities and their health status in general had worsened after COVID-19 illness, if they had remained the same, or if there had been an improvement.

Finally, the third section investigated the occupational status of the participants: participants were asked if they had been employed before COVID-19, and, if yes, they were asked if their job was full or part-time before the acute illness. Subsequently, a change in occupational status was investigated, asking participants in which type of job they were currently employed. A final question was asked about the reason for the job change, if such a change had occurred.

### 2.2. Statistical Analysis

The descriptive statistics included frequencies and percentages for categorical variables, and a calculation of mean and standard deviation (SD) for quantitative variables (age). To evaluate differences between groups for categorical variables, a univariate analysis using chi-square tests was carried out. For multivariate analysis, a logistic regression model was applied. Covariates with a *p*-value ≤ 0.25 in the univariate analysis were included in the model, according to the Hosmer and Lemeshow procedure [27]. Odds ratios (ORs) with 95% confidence intervals (95% CI) were calculated. The level of statistical significance was set at *p* ≤ 0.05.

Statistical analysis was performed with SPSS 22.0 (IBM Company, Chicago, IL, USA), software for Windows.

### 2.3. Ethical Statement

All participants were thoroughly informed about the study’s aim and purpose, and they freely gave their inform consent to participate in the study, as well as their consent for personal data processing.

The study was performed in accordance with the Helsinki declaration, the Health Insurance Portability and Accountability Act (HIPAA), and with the principles established by the 18th World Medical Assembly and all subsequent amendments and the guidelines for Good Epidemiology Practice. The study was approved by the Ethics Committee of the Università Cattolica del Sacro Cuore di Roma (ID 3777).

## 3. Results

The sample included 155 patients who completed follow-up within a year of testing positive for COVID-19. The mean age was 46.48 years (SD ± 7.302); 76 participants were males (49.7%) and 77 were females (50.3%). Concerning the behavioural habits of the interviewed population, 33 participants stated that they were smokers (21.3%), 116 that they consumed alcohol (74.8%). Most participants reported eating healthy food (90.3%) and doing physical activity (97.4%).

Out of 155 patients, 18 (11.8%) were admitted to hospital. The length of stay in hospital varied between 4 and 55 days. Before COVID-19, 151 participants (98.7%) had a full-time job, while only two participants (1.3%) had a part-time job.

The only complication reported was asthenia, occurring in 64 (41.8%) of participants.

Concerning recovery from COVID-19, 124 patients (81.0%) reported feeling fully recovered at follow-up, while 29 (19.0%) reported not feeling fully recovered. A few patients also reported performance worsening in everyday activities, 25 (16.3%) patients reported a worsening in their ability to move, 5 (3.3%) reported more difficulties performing personal care tasks, 16 (10.5%) reported an increased difficulty in performing daily activities, 37 (24.2%) reported worse pain, 71 (46.4%) had worse anxiety levels, 41 (26.8%) reported increased breathlessness, 10 (6.5%) reported vision problems, 5 (3.3%) reported hearing loss, 24 (15.7%) reported lack of focus, two (1.3%) reported language difficulties.

After suffering from COVID-19, 131 (85.6%) patients retained the same occupational status as before, while 21 (13.7%) changed. Of these, 14 workers who changed their status expressed the reasons: seven patients were on sick leave, three patients reported a loss of job due to ill health, three people reported a shortening of working hours, one person reported being fired; the other seven patients reported a change due to different reasons than those investigated or that they preferred not to answer the question.

Using a chi-square test, the correlation between employment status, hospital stay, and the ability to perform daily activities, along with other variables, was investigated; the association was considered significant for *p*-values ≤ 0.05. The only significant association with change in employment status after suffering from COVID-19 was with smoke (*p* = 0.037): 38.1% patients in the smokers group reported a change in occupational status, whilst only 18.2% of patients in the non-smokers group reported a change (Table 1).

The only statistically significant correlation for patients who reported feeling fully recovered was with hospital stay (*p* = 0.003); 113 (91.1%) patients reporting full recovery did not undergo hospital stay, while only 11 (8.9%) patients reporting full recovery underwent hospital stay (Table 2). Hospital stay was significantly (*p* = 0.025) associated with age: 53.3% of patients aged 46 years old and younger were admitted to the hospital, whilst 80.0% of patients older than 46 years of age underwent a hospital stay (Table 3). The ability to perform daily activities decreased in 56.3% of patients who underwent a hospital stay (*p* < 0.001), in 56.3% of patients who did not feel fully recovered (*p* < 0.001), and in 87.5% of patients reporting asthenia (*p* < 0.001) (Table 4). Anxiety was significantly related to gender: after suffering from COVID-19, anxiety worsened in 54.4% of female patients, but in only 36.8% of male patients (*p* = 0.028); no other significant correlation was found based on gender.

Lastly, a logistic regression was performed for the employment status. All the variables that had *p*-values < 0.250 at the chi-square test with employment status were used. The only statistically significant association was with smoking [OR 4.106, CI (1.406–11.990), *p* = 0.010] (Table 5).

## 4. Discussion

A questionnaire was administered to 155 patients with previous SARS-CoV-2 infection, to investigate health status and recovery within a year of suffering from COVID-19. Overall, we found that 19.0% of patients reported not feeling fully recovered at follow-up, and 13.7% reported a change in their job status after COVID-19. A significant (*p* ≤ 0.05) association was found between occupational status change and being a smoker, while undergoing hospital stay was associated with not feeling fully recovered and age; decreased ability to perform daily activities was associated with hospital stay, not feeling fully recovered, and asthenia; worsened anxiety was associated with female gender.

The primary outcome of this study was change in occupational status. Our results highlighted a statistically significant correlation with being a smoker; this was confirmed by the logistic regression model. This may be due to a more severe form of the COVID-19 illness occurring in smokers since they are more likely to develop respiratory infections than non-smokers, have more severe complications, as well as undergo hospital stay because of it more often [28,29,30]. Severe forms of COVID-19 in smokers could also be explained by the fact that the ACE-2 receptor, which binds the SARS-CoV-2 virus, has a higher expression in smokers [31,32]. It is important to note that, during medical surveillance, the occupational physician can establish either that the worker is fit to return to their previous job, that they are temporarily or permanently not fit to return to work, or that they may return to their job but with limitations in hours or tasks performed. While a medical surveillance decision regarding occupational status change was not investigated, a change in occupational status was significantly associated with being a smoker.

Considering the impact of smoking on the ability to return to work highlighted by our results, an important step to take would be to develop and promote smoking cessation campaigns at a company level and at a national level, to prevent more severe sequelae as well as to ensure a faster and complete recovery from COVID-19.

Furthermore, the relevance of occupational status as a quality of life factor highlights the importance of considering the ability to return to work as an important outcome in all studies addressing the recovery, or lack thereof, from COVID-19 [33]. This should be also considered when addressing the indirect impact of COVID-19 on children, as a negative impact on job activities of young adults can negatively impact their ability to care for their children [34]. It is also important to consider that our study population consisted entirely of parents: the ability to resume working after suffering from COVID-19 is even more important when considering the impact that a long absence from work or a decrease in the ability to perform daily activity may have on parents. The economic aspect also appears relevant: the impact of COVID-19 sequelae on the national healthcare system (medical services, rehabilitation, drugs consumption, etc.) is therefore worsened by the indirect cost caused by the change in occupational status. The great social and economic consequences of COVID-19 sequalae in workers and working parents highlights the need to inform the general population about the available care and rehabilitation systems in place for post COVID-19 patients, to ensure their full and prompt recovery.

This study also highlighted that 29.0% of patients not feeling fully recovered underwent a hospital stay, versus only 8.9% of patients reporting full recovery at follow-up (*p* = 0.003). These data are to be expected, as more severe form of infection may cause a slower recovery, leading to patients reporting not feeling fully recovered at follow-up. A slower recovery and higher sequelae over time after COVID-19 infection have been associated with hospital stay, and particularly intensive care unit stay, in previous studies [35,36].

Moreover, another variable associated with hospital stay was age: older (>46 year old) patients were significantly (*p* = 0.025) more likely to undergo a hospital stay, with 80% of patients above 46 years of age being hospitalized versus 53.3% of younger (≤46 year old) patients. This reinforces the knowledge that older patients tend to undergo hospital stay due to COVID-19 more often than younger people [37,38].

The ability to perform daily activities decreasing after COVID-19 was associated with undergoing a hospital stay (*p* < 0.001), not feeling fully recovered (*p* < 0.001), and asthenia (*p* < 0.001). A slower recovery in hospitalized patients has already been discussed, and it could lead to a self-reported decrease in the ability to perform daily activities in patients, which in turn would make the patient not feel fully recovered. Furthermore, asthenia is one of the most common sequelae after primary COVID-19 illness [39], and it strongly impacts daily life activities. A decrease in the ability to perform daily activities could strongly impact the ability to work, and the life of workers in general; although no significant correlation was found between the self-reported ability to perform daily activities (or the factors associated with it) and a change in occupational status, this may be due to the fact that a long enough period of time had elapsed between the primary illness and follow-up (which was performed within a year).

Anxiety worsened in 54.4% of female patients after COVID-19, but only in 36.8% of male patients (*p* = 0.028). Anxiety has been reported as a persistent problem after the acute phase of the COVID-19 illness compared to controls, and higher anxiety levels have been reported in COVID-19 patients even months after the primary infection [40,41]. Furthermore, coherently with our findings, anxiety and depression as sequelae of COVID-19 have been reported to be higher in women compared to men [41,42]. This knowledge could be important for the occupational physician to establish medical surveillance programs to monitor mental health more closely in workers after suffering from COVID-19, especially for female workers.

The results from this study highlight the need to carefully assess workers undergoing medical surveillance after suffering COVID-19. An evaluation of mental health in workers who suffered from COVID-19 and are being assessed for returning to work could prevent the worsening of anxiety caused by the acute illness and could be a useful prevention tool. Moreover, strategies to inform workers about the consequences that smoking has on their health are evermore essential, both at a company and a national level. This research could provide a starting point for policymakers, highlighting the need to integrate assessments gathered by occupational physicians, family doctors, and through the national healthcare system, to develop prevention strategies, as well as care and rehabilitation pathways, to ensure the full recovery of patients still suffering from COVID-19 sequelae.

This study has some strengths and limitations. To our knowledge, this was the first study focused on occupational health framing smoking as an important risk factor for workers who suffered from COVID-19, having a significant impact on the occupational status of the participants. Further studies may be needed, investigating the role that smoking has in the post COVID-19 condition in relation to the workers’ ability to resume their job, to more precisely frame this risk condition in the occupational setting. However, the sample was small, comprising only 155 patients, and the follow-up time was not homogeneous for the participants, not allowing for a division of the patients into follow-up groups based on the timeframe.

## 5. Conclusions

Overall, we found that 19.0% of patients reported not feeling fully recovered at follow-up, and 13.7% reported a change in their job status after COVID-19. Being a smoker has been highlighted as an important factor compromising the ability of patients to return to work after suffering from COVID-19. It also highlighted a higher prevalence of anxiety after the acute illness in female workers; furthermore, undergoing a hospital stay is associated with older age and with the patients not reporting full recovery at follow-up. This study adds to the current knowledge of COVID-19 sequelae in the occupational setting, and could be a starting point to further investigate being a smoker as a risk factor in workers not able to resume their job after the acute infection. This study’s results could be relevant in occupational health practice, reinforcing the knowledge that occupational physicians should monitor mental health more closely in the months following primary COVID-19 illness, particularly in female workers.

## Figures and Tables

**Table 1 ijerph-19-11093-t001:** Chi-square test correlating current work status (* *p* ≤ 0.05; ° *p* ≤ 0.25, included in the regression).

		Current Job		
		Same as before (n, %)	Different after COVID-19 (n, %)	*p*-Values
**Age category** **(years)**	**≤46**	59 (44.7%)	8 (36.4%)	0.465
	**>46**	73 (55.3%)	14 (63.6%)	
**Gender**	**Male**	65 (49.2%)	10 (45.5%)	0.742
	**Female**	67 (50.8%)	12 (54.5%)	
**Hospital stay**	**No**	118 (89.4%)	17 (77.3%)	0.109 °
	**Yes**	14 (10.6%)	5 (22.7%)	
**Fully recovered**	**No**	24 (18.2%)	7 (31.8%)	0.140 °
	**Yes**	108 (81.8%)	15 (68.2%)	
**Asthenia**	**No**	80 (60.6%)	10 (45.5%)	0.182 °
	**Yes**	52 (39.4%)	12 (54.5%)	
**Ability to Move**	**No**	113 (86.3%)	15 (71.4%)	0.084 °
	**Yes**	18 (13.7%)	6 (28.6%)	
**Ability to Perform Personal Care**	**No**	127 (96.2%)	22 (100.0%)	0.353
	**Yes**	5 (3.8%)	0	
**Daily Activities**	**No**	120 (91.6%)	18 (81.8%)	0.153 °
	**Yes**	11 (8.4%)	4 (18.2%)	
**Pain**	**No**	104 (78.8%)	14 (63.6%)	0.120 °
	**Yes**	28 (21.2%)	8 (36.4%)	
**Anxiety**	**No**	71 (53.8%)	13 (59.1%)	0.644
	**Yes**	61 (46.2%)	9 (40.9%)	
**Breathlessness**	**No**	96 (72.7%)	17 (77.3%)	0.655
	**Yes**	36 (27.3%)	5 (22.7%)	
**Vision Problems**	**No**	126 (95.5%)	19 (86.4%)	0.092 °
	**Yes**	6 (4.5%)	3 (13.6%)	
**Hearing Loss**	**No**	128 (97.0%)	20 (95.2%)	0.678
	**Yes**	4 (3.0%)	1 (4.8%)	
**Lack of Focus**	**No**	112 (84.8%)	18 (85.7%)	0.918
	**Yes**	20 (15.2%)	3 (14.3%)	
**Language**	**No**	130 (98.5%)	22 (100.0%)	0.561
	**Yes**	2 (1.5%)	0	
**Smoke**	**No**	108 (81.8%)	13 (61.9%)	0.037 *
	**Yes**	24 (18.2%)	8 (38.1%)	
**Alcohol**	**No**	33 (25.0%)	6 (27.3%)	0.820
	**Yes**	99 (75.0%)	16 (72.7%)	
**Healthy Food**	**No**	15 (11.4%)	0	0.096 °
	**Yes**	117 (88.6%)	22 (100.0%)	
**Physical Activity**	**No**	4 (3.0%)	0	0.408
	**Yes**	128 (97.0%)	22 (100.0%)	

**Table 2 ijerph-19-11093-t002:** Chi-square test correlating patients’ recovery with age, gender, hospital stay, change in job, asthenia, and modifiable risk factors (* *p* ≤ 0.05).

		Long COVID		
		Fully Recovered	Not Fully Recovered	*p*-Values
**Age category** **(years)**	**≤46**	58 (46.8%)	9 (29.0%)	0.075
	**>46**	66 (53.2%)	22 (71.0%)	
**Gender**	**Male**	64 (51.6%)	12 (38.7%)	0.199
	**Female**	60 (48.4%)	19 (61.3%)	
**Hospital stay**	**No**	113 (91.1%)	22 (71.0%)	0.003 *
	**Yes**	11(8.9%)	9 (29.0%)	
**Change Job**	**No**	108 (87.8%)	24 (77.4%)	0.140
	**Yes**	15 (12.2%)	7 (22.6%)	
**Asthenia**	**No**	84 (67.7%)	6 (19.4%)	0.182
	**Yes**	40 (32.3%)	25 (80.6%)	
**Smoke**	**No**	98 (79.0%)	23 (76.7%)	0.777
	**Yes**	26 (21.0%)	7 (23.3%)	
**Alcohol**	**No**	29 (23.4%)	10 (32.3%)	0.820
	**Yes**	95 (76.6%)	21 (67.7%)	
**Healthy Food**	**No**	13 (10.5%)	2 (6.5%)	0.497
	**Yes**	111 (89.5%)	29 (93.5%)	
**Physical Activity**	**No**	4 (3.2%)	0	0.408
	**Yes**	120 (96.8%)	31 (100.0%)	

**Table 3 ijerph-19-11093-t003:** Chi-square test correlating age, gender, and modifiable risk factors, with hospital stay (* *p* ≤ 0.05).

		Hospital Stay		
		No (n, %)	Yes (n, %)	*p*-Values
**Age category** **(years)**	**≤46**	63 (46.7%)	72 (53.3%)	0.025 *
	**>46**	4 (20.0%)	16 (80.0%)	
**Gender**	**Male**	63 (46.7%)	13 (65.0%)	0.126
	**Female**	72 (53.3%)	7 (35.0%)	
**Alcohol**	**No**	37 (27.4%)	2 (10.0%)	0.094
	**Yes**	98 (72.6%)	18 (90.0%)	
**Healthy Food**	**No**	15 (11.1%)	0	0.117
	**Yes**	120 (88.9%)	20 (100.0%)	
**Physical activity**	**No**	0 (3.0%)	0	0.435
	**Yes**	131 (97.0%)	20 (100.0%)	
**Smoke**	**No**	107 (79.9%)	14 (70.0%)	0.317
	**Yes**	27 (20.1%)	6 (30.0%)	

**Table 4 ijerph-19-11093-t004:** Significant chi-square tests correlating ability to perform daily activities with hospital stay, feeling fully recovered and asthenia (* *p* ≤ 0.05).

		Ability to Perform Daily Activities		
		Same as before (n, %)	Decreased after COVID-19 (n, %)	*p*-Values
**Hospital stay**	**No**	127 (92.0%)	7 (43.8%)	<0.001 *
	**Yes**	11 (8.0%)	9 (56.3%)	
**Fully recovered**	**No**	22 (15.9%)	9 (56.3%)	<0.001 *
	**Yes**	116 (84.1%)	7 (43.8%)	
**Asthenia**	**No**	88 (63.8%)	2 (12.5%)	<0.001 *
	**Yes**	50 (36.2%)	14 (87.5%)	

**Table 5 ijerph-19-11093-t005:** Logistic regression analysis for current employment (* *p* ≤ 0.05).

		Current Job		
		OR	CI [0–95%]	*p*-Value
**Age**		0.987	0.919–1.060	0.716
**Gender**	Male	1		
	Female	1.125	0.397–3.189	0.825
**Hospital Stay**	No	1		
	Yes	1.674	0.420–6.671	0.465
**Fully Recovered**	No	1		
	Yes	1.101	0.264–4.598	0.895
**Asthenia**	No	1		
	Yes	1.428	0.404–5.050	0.581
**Ability to move**	Same as before	1		
	Worsened	1.212	0.227–6.463	0.822
**Ability to Perform Daily Activities**	Same as before	1		
	Worsened	1.367	0.244–7.661	0.722
**Pain**	Same as before	1		
	Worsened	2.886	0.967–8.607	0.057
**Vision Problems**	Same as before	1		
	Worsened	3.031	0.611–15.033	0.175
**Smoke**	No	1		
	Yes	4.106	1.406–11.990	0.010 *

## Data Availability

Data from this research is available upon reasonable request.

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
