# Peer review of "Post-Acute COVID-19 Sequelae in a Working Population at One Year Follow-Up: A Wide Range of Impacts from an Italian Sample"

_ijerph, 2022, doi:10.3390/ijerph191711093_

Round 1

Reviewer 1 Report

Dear authors,

The manuscript deals with a relevant topic, but language and content improvement are necessary.

Overwall: to improve English writing. There are confusing parts, which lead to misinterpretation due to the translation errors.

To improve objectives writing to beyond broadening the evidence, but exactly what is being analyzed. There is no consistency between title, objective and the study population.

There is a need to detail methodological steps, and not to mention that they are writen in another study.

Discussion: there is a lot of repetition of the data already presented in the results. Furthermore, considering that there is an intention to present new evidence, what progress is there? For the discussion is only based on what previous references demonstrate.

There is a need to deepen the discussion and present new inferences based on the results.

Author Response

Dear reviewer,

Thank you so much for reviewing our manuscript and for your valuable input.

Regarding your concerns:

An extensive English revision has been performed, thank you for suggesting this to us. You can find all the changes in track change mode in the manuscript.

The objective of the study has been amended as follows: “The aim of this study is to evaluate sequelae of COVID-19 in a population of workers tested positive for COVID-19, with a follow-up within one year of the acute illness, and analyzing the possible association between COVID-19 sequelae and changes in the workers' occupational status.

Concerning your comment on methodology, we have mentioned the ISARIC because the protocol and questionnaire were developed for that study, but the sentence was not clear. Thank you for pointing it out, we have rephrased as follows: “The questionnaire used in this study was developed for the ISARIC Global COVID-19 follow up protocol for adults [21,22] and associated standardized data collection forms [23–26].” After the paragraph we proceed to describe the questionnaire more in depth.

Thank you so much for your suggestion concerning the discussion section; it has been improved adding paragraphs to further discuss our findings on being a smoker, the impact of COVID-19 on parents/families, and what this study could highlight for policymakers and other researchers.

Reviewer 2 Report

Dear Authors,

thank you very much for this paper. However, I have some concerns

I have a few suggestions for improvement.

a) major

- Page 1, line 23: It is recommended to reformulate the objective, the verb to study is not measurable.

- Page 2, lines 83-97: I recommend that you make changes to make this paragraph better understood. For example, in lines 93-94 it says "The data was collected in a pediatric unit of a large hospital in Rome, Italy" and it already mentions it before. I recommend that you use shorter and simpler sentences.

- Page 2, lines 78-81: It is advisable to write the objectives more clearly so that the results and discussion section can be better understood.

- Page 5, Table 1: In the gender variable, when adding the feminine ones of the two columns it is 75 and in the text appears "77 were females" (page 4, line 148). With the masculine ones when adding the two masculine columns it is 79 and in the text appears "76 participants were males" (pages 3-4, lines: 147-148). When adding female and male (75+79) you get 154 but you said in the text "A sample of 153..." (page 2, line 92).

- Page 5, Table 1; page 6, Table 2 and Table 3; page 7, Table 4: If n=153, why is the sum of each of the different variables not equal to that number? If it was due to non-response on those items, please indicate this in Methods. I recommend reviewing n and % because some don't add up.

-Page 7-9, discussion section: In the Discussion, it is more of a repetition of the main results and international literature on the aspects, rather than a focussed (research question answering) discussion of the main aspects.

b) minor

- Page 1: It would be advisable for the study design to appear in the title or abstract.

- Page 1, line 25: “(using chi-square tests)” delete using.

- Page 1, line 29: “[1,406 – 11,990]” change by [1.406 – 11.990]

- Page 3, lines 128-129: When it says “The study outcomes were: worsening of  the ability to carry out normal day-to-day activities and changing occupational status" Why does it appear in the methodology section? In my opinion, this should appear in the results section.

- Page 3, line 132: When it says “Hosmer and Lemeshow procedure (REF)”, missing to add the reference.

- Page 3, line 132: “(95% CIs)” change by (95% CI).

- Page 5, Table 1: The category "Yes" is repeated twice in the asthenia variable.

Author Response

Dear reviewer,

Thank you so much for your valuable input and for reviewing our manuscript.

Regarding your major concerns:

The objective of the study has been changed to: “The aim of this study is to evaluate sequelae of COVID-19 in a population of workers tested positive for COVID-19, with a follow-up within one year of the acute illness, and analyzing the possible association between COVID-19 sequelae and changes in the workers' occupational status.” We have shortened our sentences and avoided repetition, clarifying the objective as per your valuable suggestion.

Thank you so much for your comment on the tables n and %; we have double checked everything and have realized there was a discrepancy between descriptive and analytic analyses, we have amended the sample size to the actual total (155) and have added this sentence regarding missing values: “For unanswered questions the data was considered missing and that value was excluded from the statistical analyses.”

Concerning the discussion: thank you so much for your suggestion, this section has been improved by adding paragraphs to further discuss our findings on being a smoker, the impact of COVID-19 on parents/families, and what this study could highlight for policymakers and other researchers.

Regarding your minor concerns:

Thank you for pointing this out to us, we have added the study design to the abstract: “In this retrospective cohort study […]”. We have deleted the term “using” and have amended the punctuation before decimals.

We have removed the outcomes from the method section, thank you for your input.

The Hosmer and Lemeshow procedure’s reference was added; “CIs” was amended to “CI”; the table’s categories have been corrected.

Round 2

Reviewer 1 Report

Dear authors,

The manuscritp contains a good content and the new version has a incresed quality with writing review.

Best Regards.

Reviewer 2 Report

Dear Authors,

The changes made have improved his work.

Thank you very much for this paper.